

# Land surface temperature trends derived from Landsat imagery in the Swiss Alps

Deniz Tobias Gök[1], Dirk Scherler[1,2], Hendrik Wulf[3]

[1] GFZ German Research Centre for Geosciences; D-14473 Potsdam, Germany.
  [2] Institute of Geographical Sciences, Freie Universität Berlin; D-14195 Berlin, Germany.
  [3] Remote Sensing Laboratories, University of Zurich; CH-8057 Zurich, Switzerland.

Correspondence to: Deniz Tobias Gök (d_goek@gfz-potsdam.de)

**Abstract.** The warming of high mountain regions caused by climate change is leading to glacier retreat, decreasing snow
cover, and thawing permafrost, which has far-reaching effects on ecosystems and societies. Landsat Collection 2 provides multi-decadal land surface temperature (LST) data, principally suited for large-scale monitoring at high spatial resolution. In this study, we assess the potential to extract LST trends using Landsat 5, 7, and 8 time series. We conduct a comprehensive comparison of both LST and LST trends with data from 119 ground stations of the IMIS network, located at high elevations in the Swiss Alps. The direct comparison of Landsat and IMIS LST yields robust satellite data with a mean accuracy and
precision of 0.26 K and 4.68 K, respectively. For LST trends derived from a 22.6-year record length, as imposed by the IMIS data, we obtain a mean accuracy and precision of -0.02 K yr⁻¹ and 0.13 K yr⁻¹, respectively. However, we find that Landsat-LST trends are biased due to unstable diurnal acquisition times, especially for Landsat 5 and 7. Consequently, LST trend maps derived from the 38.5-year Landsat data exhibit systematic variations with topographic slope and aspect that we attribute to changes in direct shortwave radiation between different acquisition times. We discuss the origin of the magnitude and spatial
variation of the LST trend bias in comparison with modelled changes in direct shortwave radiation and propose a simple approach to estimate the LST trend bias. After correcting for the LST trend bias, remaining LST trend values average between 0.07 and 0.10 K yr⁻¹. Further, the comparison of Landsat- and IMIS-derived LST trends suggests the existence of a clear-sky bias, with an average value of 0.027 K yr⁻¹. Despite these challenges, we conclude that Landsat LST data offer valuable high-resolution records of spatial and temporal LST variations in mountainous terrain. In particular, changes in the mountain
cryosphere such as glacier retreat, glacier debris cover evolution and changes in snow cover, are preserved in the LST trends and potentially contribute to improved prediction of permafrost temperatures with large spatial coverage. Our study highlights the significance of understanding and addressing biases in LST trends for reliable monitoring in such challenging terrains.

## 1 Introduction

The Earth's surface temperature, at the land-atmosphere interface, is a key parameter of the surface energy budget
and influences a range of biological, chemical, and physical processes within the critical zone (e.g., Brantley et al., 2007). It





reflects both climate change and land surface processes and is defined as an essential climate variable by the World Meteorological Organisation (Bojinski et al., 2014). Increasing surface temperature is expected to have a severe adverse impact on ecosystems, human health, and infrastructure (IPCC, 2023). With time, surface warming propagates to greater depths, resulting in additional changes. High mountain regions that often host glaciers, snow cover, and permafrost, are particularly

sensitive to increasing temperatures. Where mean annual ground temperatures rise to above 0°C, permafrost thaws, thereby destabilizing steep hillslopes (Gruber and Haeberli, 2007; Huggel, 2009; Allen et al., 2009). Indeed, increased rockfall activity and several recent significant slope failures in the European Alps (Gruber et al., 2004; Harris et al., 2009; Walter et al., 2020) have been linked to permafrost thaw. Such catastrophic events pose serious hazards to both people and infrastructure in numerous mountain ranges on Earth. Monitoring Earth's surface temperature and its spatiotemporal variation, therefore,

significantly contributes to the improved prediction of the impacts of Global Warming. Ground-based instrumental monitoring of the surface temperature, however, is laborious and difficult to implement over large regions and in remote mountainous areas with steep hillslopes. Therefore, the spatial coverage of station-based surface temperature data is limited, especially when it comes to long-term records.

Satellite platforms equipped with thermal infrared sensors, allow measuring the land surface temperature (LST) at a

range of spatial and temporal resolutions and have long been used in a variety of research fields (Li et al., 2013, Hulley et al., 2019, Reiners et al., 2023; Li et al., 2023). Temporal LST analysis for climate change studies or environmental monitoring requires multi-decadal time series data, which often encounters the challenge of maintaining the temporal coherence of the thermal data (Kuenzer and Dech, 2013). Many LST studies rely on data from the Moderate Resolution Imaging Spectroradiometer (MODIS) sensor onboard the Terra and Aqua satellites (Reiners et al., 2023). MODIS LST records are

temporally consistent (Hulley and Hook, 2011) and LST trends have been recently derived globally (Sobrino et al., 2020). However, the relatively coarse spatial resolution of the thermal bands (1000 m) restricts the applicability of MODIS LST in high mountainous regions, where the steep terrain results in large spatial gradients in surface temperatures. In addition to altitudinal gradients in temperature, due to the decreasing air temperature, temperature variations also exist in response to variable exposition to the sun.

As the robustness of trends increases with longer time series, LST records from the Advanced Very-High-Resolution Radiometer (AVHRR) and the Landsat Program, are particularly useful for this purpose (Prata, 1994; Gutman and Masek, 2012). Both suffer, although in different manner, from orbital drift effects, causing the acquisition time to vary over time (Julien and Sobrino, 2022, Zhang and Roy, 2016). Orbital drift corrections for AVHRR LST time series are continuously developed (e.g., Gutman et al., 1999; Mao and Treadon, 2013; Dech et al., 2021, Julien and Sobrino, 2022), as the daily

temporal resolution allows unique insights into long-term dynamics of LST. Landsat, with its lower temporal but higher spatial resolution, has so far been underutilized for time series analysis (Fu and Weng, 2015). The recently released Landsat Collection 2, with improved radiometric calibration and geolocation information (Crawford et al., 2023), provides consistently generated LST data (Malakar, 2018). Landsat-derived LST time series therefore present a unique opportunity to explore the dynamics of high mountain landscapes in response to climate change and human land cover modifications.



For instance, recently published LST trends of glacier surfaces in High Mountain Asia show enhanced surface
warming trends due to supraglacial debris cover and its expansion (Ren et al., 2024). Spatial patterns in LST trends are also
expected in areas of seasonal snow cover. Especially at altitudes near the 0 °C isotherm, small changes in air temperature can
have a significant impact on snow cover (Pepin and Lundquist et al., 2008). Observations show that in the European Alps snow
cover declines in extent, duration and depth (Matiu et al., 2021) with vegetation expanding into higher elevations and thus

changing the surface albedo (Rumpf et al., 2022). Furthermore, because mountain permafrost temperatures vary in response
to changes in air temperature and snow cover (Smith et al., 2022), spatial patterns in LST and LST trends have the potential to
inform about expected spatial variations in permafrost temperature, depth and extent. Despite sufficiently long records and the
high spatial resolution of Landsat observations, deriving LST trends is complicated as acquisition times have changed by up
to 1 h (Roy et al., 2020), due to orbit changes over the last decades (Zhang and Roy, 2016).

Here, we explore the opportunities of monitoring LST trends in steep mountainous regions using Landsat Collection
2. We first assessed the reliability of Landsat-derived LST and LST trends by comparison with ground observations from the
Intercantonal Measurement and Information System (IMIS) network, which provides comparable radiometric surface
temperatures at high-elevation sites across the Swiss Alps (Figure 1). We then calculated spatially distributed LST trends and
identify a spatially variable bias that we associate with orbital drift of the satellites. We analyse the magnitude and spatial

variation of this bias and present a simple approach to correct for it. Additionally, we address issues related to the clear-sky
bias and explore opportunities and limitations for studying cryosphere changes using the corrected Landsat LST trends.

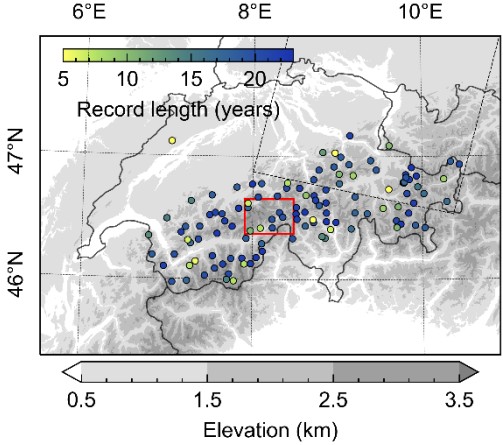

**Figure 1. Intercantonal Measurement and Information System (IMIS) network of automated weather stations distributed across the
Swiss Alps. The station data provide radiometric surface temperatures at 30-minute intervals with varying time spans, indicated by
inset color. The red rectangle identifies the upper Rhone Valley shown in Figure 7. The black dashed rectangle indicates the Landsat
footprint at path 194 and row 27, referred to in Figure 2.**



## 2 Materials and Methods

### 2.1 Landsat-derived LST

Landsat Collection 2 (C2) - Level-2 Science Products provide multi-decadal observational remote sensing data that
is geometrically and radiometrically consistent and has harmonized quality assessment bands (Dwyer et al., 2018). We used
the Google Earth Engine (GEE) to analyze LST data (Malakar et al., 2018) from Landsat 5 Thematic Mapper (TM), Landsat
7 Enhanced Thematic Mapper Plus (ETM+) and Landsat 8 Thermal Infrared Sensor (TIR) (hereafter LT05, LE07, LC08)
covering a timespan from 1984 to 2022. The native spatial resolutions of LT05 (120 m), LE07 (60 m) and LC08 (100 m) have
been resampled in Collection 2 to 30 m, which is the spatial resolution that we used in our study.

The Landsat C2 LST calculation is based on the single-channel algorithm (Malakar et al., 2018) that relies only on
one thermal infrared band and which has been widely used to retrieve LST from Landsat data (Jiménez-Muñoz and Sobrino,
2003; Cook et al., 2014). The conversion of at-sensor radiometric temperature to LST requires an atmospheric correction and
knowledge of the surface emissivity. The atmospheric correction in the Landsat C2 LST calculation is based on the total
column water vapor derived from NCEP atmospheric reanalysis data (Kalnay et al., 1996). Mean emissivity estimates over the
time period 2000-2008 are based on the Advanced Spaceborne Thermal Emission and Reflection Radiometer Global
Emissivity Dataset (ASTER GED) (Hulley et al., 2015) and temporally adjusted using Landsat-derived NDVI and NDSI
(Normalized difference snow index). Inspection of the ASTER GED reveals several artifacts, which appear to align with
artefacts in the Landsat LST data. To avoid erroneous LST data and mask out clouds in the Landsat images, we applied several
filters and masks that we describe in more detail in section 0.

The scene acquisition time of Landsat for the Swiss Alps lies mostly between 09:30 and 10:30 UTC. Figure 2 shows
the acquisition times from the different Landsat sensors during the study period. Whereas LC08 has a relatively stable
acquisition time, LE07 shows slightly continuous drift before and strong drift after about 2018, due to depleted onboard fuel
resources (Qiu et al., 2021). LT05 on the other hand shows both sporadic and continuous orbit changes that lead to significant
variations in acquisition time (Zhang and Roy, 2016). Although orbit variations are often due to sporadic orbit keeping
maneuvers, a gradual increase in overpass times is evident too (Roy et al., 2020). When fitting a linear model to all satellites
together (but excluding LE07 data after 2018 due to strong orbital decay), the acquisition time has increased approximately
from 9:29 in 1984 to 10:16 in 2022 (Figure 2, dotted line). We expect that LST trends derived from the 38.5-year time series
are likely biased by the progressively delayed acquisition times, probably towards more positive values, due to gradual
warming of the land surface in the morning. Because different acquisition times also lead to geometric changes in the sun-
target-sensor configuration, this bias may additionally vary with slope and aspect of the topography. We describe our approach
to analyze this issue in section 0.





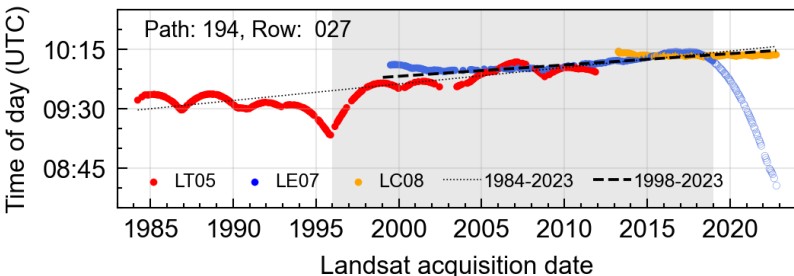

**Figure 2. Acquisition times (UTC) of Landsat LT05 (red), LE07 (blue), and LC08 (orange) at path 194 and row 027. LE07's noticeable orbital drift after 2018 (hollow blue circles), causes a significant shift in revisit timing and has been excluded from the**
**analysis. Linear regression lines (dotted and dashed) depict acquisition time trends, with and without abrupt LT05 orbit changes prior to 2000. The gray-shaded area indicates the time period for which IMIS station data exists, although with variable record length.**

**2.2 IMIS-derived LST**

We evaluated the Landsat-derived LST data by comparing them with in situ surface temperature measurements from
automated weather stations of the IMIS network. The IMIS network was set up by the Swiss Federal Institute for Snow and Avalanche Research (SLF) and consists of 186 stations distributed across the Swiss Alps. We used a subset of 119 stations (Table A.1) that record radiometric surface temperature in 30 min intervals. The record length per station varies, with the longest record covering a period from 1996 to 2019 (Figure 1). The IMIS stations are located between 1258 m and 2953 m elevation above sea level and are usually installed on flat to gentle sloping ground. As the stations are primarily used for snow
monitoring, the reported surface temperature is calibrated using an emissivity of 0.98 (for snow), which may thus introduce a bias towards colder temperatures during snow-free conditions. Because the transition between snow-covered and snow-free conditions cannot be unambiguously determined based on the IMIS data alone, and because of unknown actual emissivity values of the ground surface, we refrained from efforts to correct this bias. For a surface temperature of 15 °C, a change in emissivity of 0.01 would result in a temperature change of 0.73 K (Kuenzer and Dech, 2013). This bias decreases for lower
LST values. Despite potential measurement deviations under snow-free conditions, the IMIS stations measure radiometric surface temperatures and are thus well suited to compare with Landsat derived LST. Additionally, the high temporal resolution of the IMIS data allows to compare LST clear-sky and cloudy-sky conditions using the Landsat overpass times. We expect the large difference in spatial resolution to introduce additional uncertainty as Landsat most likely provides a mixed-pixel signal of varying LSTs, compared to the IMIS data.

**2.3 LST processing and trend estimation**

For the studied period and the chosen Landsat sensors, we obtained for each 30-m pixel in the co-registered image collection several hundred LST observations scattered across different times of a year. We used a harmonic model including a linear trend (Eq. 1) to perform an ordinary least squares regression (Shumway and Stoffer, 2016; Fu and Weng, 2015) on the



LST time series data in order to estimate (1) the mean annual LST (MALST), (2) the annual LST amplitude, (3) the long-term
LST trend and (4) the phase shift:

$$LST_t = \beta_0 + \beta_1 t + A\cos(2\pi\omega t - \varphi)$$    Eq. 1

where $\beta_0$ is the mean annual LST (K), $\beta_1$ is the slope (K yr$^{-1}$) of the linear trend, $t$ is the time in years, $A$ is the amplitude (K),
$\omega$ is the frequency (equal to one for one cycle per year) and $\varphi$ is the phase. The harmonic term can be decomposed into a sine
and a cosine term, and thus Eq. 1 is linearized to:

$$LST_t = \beta_0 + \beta_1 t + \beta_2 \cos(2\pi\omega t) + \beta_3 \sin(2\pi\omega t)$$    Eq. 2

Where $\beta_2$ and $\beta_3$ are the newly introduced coefficients that are equal to $A\cos(\varphi)$ and $A\sin(\varphi)$, respectively. GEE allows
ordinary least squares regression of Eq. 2 and thus the determination of the four coefficients $\beta_0$ to $\beta_3$. We acknowledge that
LST time series may contain abrupt changes due to land cover change, for example, which may not be well captured by a
linear model (Zhu and Woodcock, 2012). Different approaches have been proposed to detect such changes and simultaneously
obtain trend values (see the recent review by Li et al., 2022). However, the change detection approaches currently available in
GEE are more limited (Kennedy et al., 2010; Zhu and Woodcock, 2012) and as we will show later, the segmentation of the
time series affects our ability to account for LST trend bias due to orbital drift.

Prior to fitting Eq. 2 to the Landsat LST data, we implemented filters to mask (1) cloud-contaminated pixels and (2)
duplicate LST observations with the same date that result from along-track overlapping Landsat scenes. Cloud masking was
done using the Landsat C2 Pixel Quality Assessment Band (QA) cloud flag (Dwyer et al., 2018; Zhu and Woodcock, 2012).
Although the cloud flag of the QA band provides good accuracy (Foga et al., 2017), bright surfaces such as snow and ice in
high mountain settings, can still be challenging. Predominantly in LT05 data, we find extremely cold LST values, which are
likely clouds that were not captured by the cloud detection algorithm. To overcome this issue, we applied an additional filter
that masks outliers, by applying a threshold to the residuals between modeled and observed LST. We first calculated the $\beta$
coefficients on the cloud-filtered data, including potential outliers missed by the QA cloud flag, and then uploaded them to
GEE. In a second step, we predicted for each Landsat acquisition time the corresponding LST using the uploaded $\beta$ coefficients
(Eq. 2) and applied a threshold of +/-30 K to the residuals to mask extreme LST values that might otherwise bias the LST trend
(cf., Weng and Fu, 2014). The procedure was applied to the complete Landsat time series data. Figure 3 shows an exemplary
LST time series from each sensor, the harmonic model with linear trend, the residuals, and the filtered outliers at the location
of IMIS station AMD2. Identical figures from all IMIS locations can be found in the supplement file B.



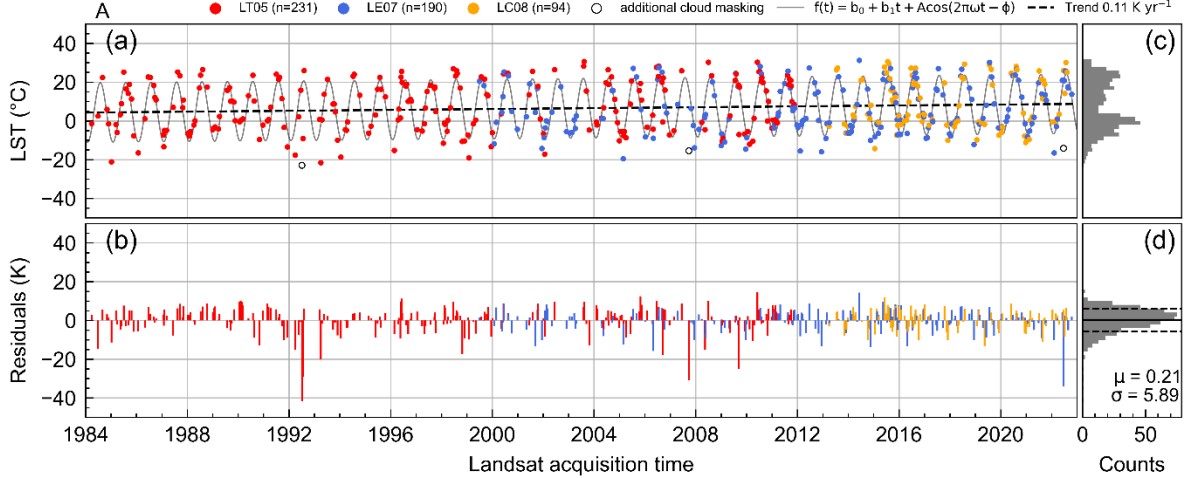

**Figure 3. Time series of Landsat LT05 (red), LE07 (blue) and LC08 (orange)-derived Land Surface Temperature (LST) at location 47.17° N, 9.15° E (IMIS station AMD2). The harmonic model (solid sinusoidal line) was derived by least squared regression including linear trend component (dashed line). Outliers (hollow circles) were detected by applying a threshold of 30 K to the (b) residuals and removed from further analysis. Panel (c) and (d) show the distribution of the LST and residuals respectively.**

To assess the reliability of the Landsat-derived LSTs and LST trends, we compared them with LST data derived from the IMIS network. We first extracted the Landsat LST time series at the locations of the IMIS stations. As IMIS records are only available in 30-minute intervals, we linearly interpolated LSTs at the Landsat acquisition times to obtain comparable LST time series of equal length. Based on Eq. 2, we derived the mean annual LST, the LST amplitude, the phase of the harmonic oscillation, and LST trends for both datasets. We further assessed the sensitivity of LST trends to the LT05, LE07, and LC08 sensors by comparing data from each sensor with the corresponding IMIS LST data, where the observation periods overlap. Because the temporal overlap of the individual Landsat sensors and the IMIS data varies, this comparison also results in different record lengths.

We used student's *t*-test to draw statistical inference for the regression slope and evaluate the significance of LST trends (Muro et al., 2018). Pixels with non-significant trends (p-values < 0.05) were flagged. Note that the comparison of LST trends between Landsat and IMIS data, as well as the spatial analysis of LST trends in relation to the LST trend bias is based on all trend data and does not require statistical significance of trend values.

**2.4 LST trend bias analysis**

We expect the LST trend to be biased due to the variations in acquisition time caused by orbital change of the satellites (Figure 2). Within the 47 minutes time difference in image acquisition between the beginning and the end of the 38.5-year Landsat observation period, the sun's position and thus also the solar zenith angle changes notably, modifying the amount of incoming shortwave radiation received by the surface. In mountainous terrain with variably steep and exposed topography, we expect this effect to be spatially non-uniform. Based on the fitted linear model of the acquisition time, we analyzed changes in



the incoming direct solar radiation ($\Delta S_{in}$) for the Swiss Alps using the "insol" functional library (Corripio, 2003). We studied the relationship of LST trends and $\Delta S_{in}$ with topography by aggregating mean values for 2° slope and 10° aspect sections

derived from the 10-m resolution Copernicus digital elevation model (Copernicus DEM, 2022). Prior to averaging LST trends we excluded glaciers and recently deglaciated areas using a mask based on glacier outlines from the Randolph Glacier Inventory V6 (RGI Consortium, 2017), which we expanded by 10 pixels in the 30-m resolution LST trend images. Additionally, we excluded all regions below 1700 m elevation, which are likely influenced by anthropogenic land cover changes (Rumpf et al., 2022).

**2.5 Validation metrics**

The LST data used in this study, obtained from the Landsat C2 archive, is based on three different sensors (LT05, LE07 and LC08) and auxiliary datasets such as the ASTER GED and NCEP reanalysis data. Since all these datasets have their limitations, it is important to validate LST data to ensure its accuracy and reliability. We compared the Landsat-derived LST with in situ LST measurements from the IMIS stations at the Landsat acquisition time. We followed the "Land Surface

Temperature Product Validation Best Practice Protocol" (Guillevic et al., 2018) by using metrics of accuracy, precision and uncertainty for reporting LST validation results. The accuracy ($\mu$), as a measure of the systematic error/bias, is given by the arithmetic mean of the difference between the satellite derived LST and the in situ measured reference LST ($\Delta LST_{ref}$). The precision ($\sigma$) describes the spread of the LST around the expected value ($\Delta LST_{ref}$) and can be approximated by the standard deviation. The uncertainty is given by the Root Mean Square Error (RMSE) and describes the dispersion of the LST values.

Because the accuracy and precision of LST data can be strongly affected by outliers, we also report the median of the $\Delta LST_{ref}$ for the accuracy and the median absolute deviation of the residuals for the precision as additional validation metrics (Guillevic et al., 2018). We apply these validation metrics to both the LST data and the LST trends. We emphasize that in our study the term "validation" may be slightly misleading as it suggests that the ground-based IMIS measurements provide the correct LST values. However, we note that even the IMIS data is most likely biased during snow-free conditions (see section 2.2) and

subject to measurement uncertainties. In addition, the different footprint of the ground- (~10 cm) and space-borne (~10-100 m) measurements allow for deviations due to spatial heterogeneity in LST. We will come back to this issue in our discussion. Nevertheless, we argue that the comparison of these data sets is a valuable effort and that consistency between both temperature measurements provides confidence.

**3 Results**

**3.1 LST comparison**

For comparing Landsat-derived LST with ground-based LST from the IMIS network, we interpolated IMIS LST's at the Landsat acquisition time. In total 44981 Landsat observations are available for comparison with IMIS observations. The



LST data from all three Landsat sensors are scattered about the 1:1 line in comparison with the IMIS data (Figure 4 a-d). At around 0°C IMIS LST, the spread in Landsat-derived LST is the highest, which is likely related to differences in spatial resolution and the presence or absence of snow cover in the different measurement areas. It furthermore appears that LSTs derived from each Landsat sensor tend to be slightly warmer for LSTs above 0 °C compared to those below 0 °C. Mean- and median-based metrics of accuracy (μ), precision (σ) and uncertainty (RMSE) between Landsat and IMIS LST for each sensor and the entire time series, as shown in Figure 4 and Table 1. The accuracy (μ) ranges from +0.05 K (LC08) to +0.45 K (LE07) and indicates a slight positive bias. The precision (σ) ranges from 4.09 K (LE07) to 6.13 K (LT05). Considering data from all three sensors together (Figure 4d), the accuracy is +0.26 K, the precision is 4.69 K and the uncertainty is 4.7 K (Table 1). Considering median values, the precision improves but the accuracy deteriorates.

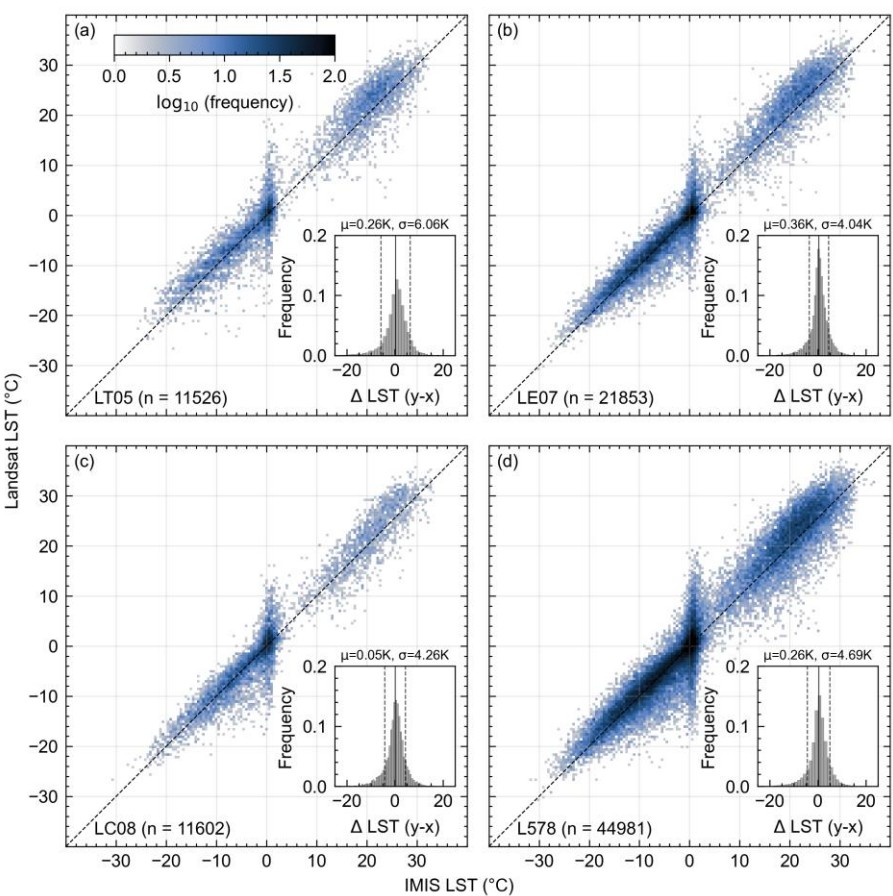

**Figure 4. Comparison of Landsat-derived Land Surface Temperature (LST) with IMIS LST for sensors (a) Thematic Mapper (LT05), (b) Enhanced Thematic Mapper Plus (LE07), (c) Thermal Infrared Sensor (LC08) and (d) LT05, LE07, and LC08 together (L578). Colors denote the number of data points by decadal logarithm. Inset figures show histograms of LST residuals: ΔLST = Landsat LST-IMIS LST.**

**Table 1. Validation metrics of Landsat-derived LST in comparison with IMIS-derived LST.**



| Quantity | Symbol | Unit | LT05 | LE07 | LC08 | L578 |
|---|---|---|---|---|---|---|
| Accuracy (mean/median) | μ | K | 0.26/0.72 | 0.36/0.5 | 0.05/0.31 | 0.26/0.5 |
| Precision (mean/median) | σ | K | 6.06/2.47 | 4.04/1.70 | 4.26/2.05 | 4.69/2.01 |
| Uncertainty (RMSE) | RMSE | K | 6.07 | 4.06 | 4.26 | 4.7 |
| Sample number | n | - | 11526 | 21853 | 11602 | 44981 |

## 3.2 LST trend comparison

We also compared Landsat-derived LST trends with trends derived from IMIS LST data interpolated at Landsat observation times, for each sensor as well as the complete time series (Figure 5, Table 2). We excluded stations with record lengths of less than 5 years. Short time series result from different temporal overlaps between the IMIS records and Landsat sensors, in particular LT05 and LC08 (Figure 5 a, c). These show large scatter about the 1:1 line compared to trends derived from longer time series, resulting in relatively large uncertainties (Table 2). Therefore, amongst the different sensors, LE07

provides the most reliable results (Figure 5b), with better accuracy and precision (Table 2), due to the large temporal overlap with the IMIS data. Consequently, our comparison of trends derived from all sensors with IMIS-derived LST trends (Figure 5d) is primarily dominated by LE07. Considering data from all three sensors together, the accuracy is -0.02 K yr$^{-1}$ and the precision is 0.13 K yr$^{-1}$, improving considerably when referring to median values.



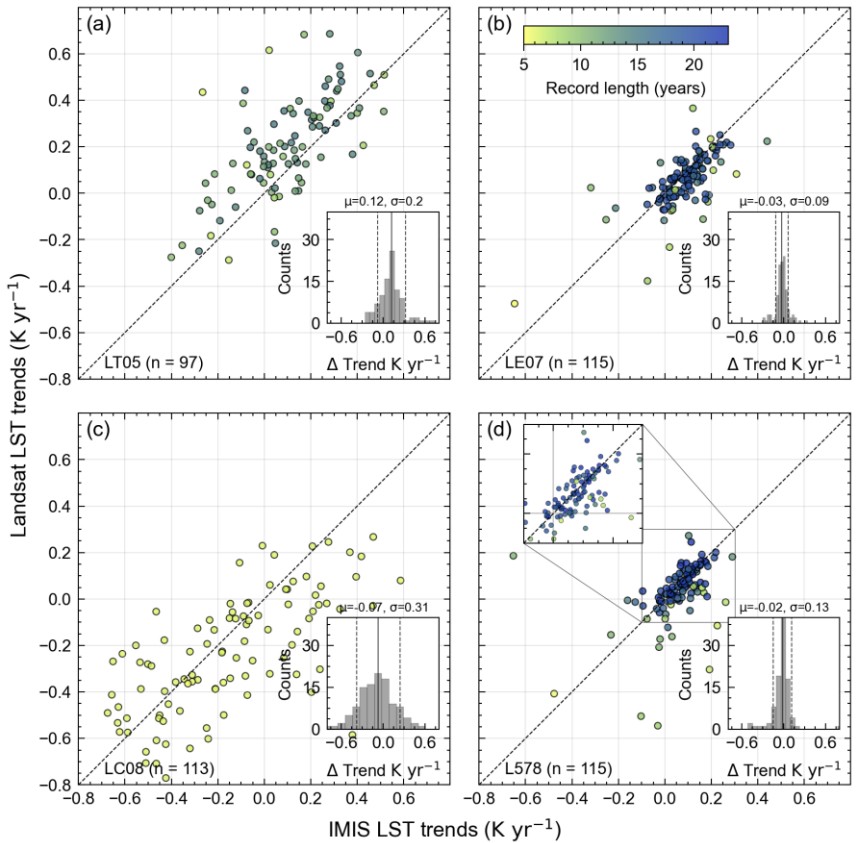

**Figure 5. Comparison of Landsat-derived Land Surface Temperature (LST) trends with IMIS LST trends for sensors (a) Thematic Mapper (LT05), (b) Enhanced Thematic Mapper Plus (LE07), (c) Thermal Infrared Sensor (LC08) and (d) LT05, LE07, and LC08 together (L578). Stations with a record length (marker color) of less than five years have been omitted. Trend residuals (Landsat LST trends – IMIS LST trends) together with the accuracy (μ) and precision (σ) values are shown in the inset histograms. Note the strong impact of record length on the comparison of LST trends.**

**Table 2. Validation metrics of Landsat-derived LST trends in comparisons with IMIS-derived LST trends.**

| Quantity | Symbol | Unit | LT05 | LE07 | LC08 | L578 |
|---|---|---|---|---|---|---|
| Accuracy (mean/median) | μ | K yr$^{-1}$ | 0.12/ 0.11 | -0.03/ -0.02 | -0.07/ -0.06 | -0.02/ -0.01 |
| Precision (mean/median) | σ | K yr$^{-1}$ | 0.20/ 0.13 | 0.09/ 0.05 | 0.31/ 0.19 | 0.13/ 0.04 |
| Uncertainty (RMSE) | RMSE | K yr$^{-1}$ | 0.23 | 0.10 | 0.31 | 0.13 |
| Sample number | n | - | 97 | 115 | 113 | 115 |

## 3.3 Spatiotemporal variations of LST

We applied Eq. 2 to the Landsat LST time series (LT05, LE07 and LC08) across Switzerland using GEE. The model results are presented as maps of the mean annual land surface temperature (MALST), the LST amplitude, the phase of the harmonic oscillation and the LST trend in Figure 6, with a focus on the upper Rhone Valley shown in Figure 7. The presented





MALST values are for the year 2000 and range from -25°C to +25°C. We observe consistently the highest MALST values at low elevations and the lowest at high elevations, where snow- and ice-covered areas range from 0°C to -20°C. As seen in the detailed map in Figure 7a, MALST values show reasonable spatial variations with terrain aspect and no significant processing artefacts are present. East-facing slopes consistently display higher MALST compared to west-facing ones, which aligns with expectations due to the late morning overpass of the Landsat satellites (Figure 7a). Data gaps, which are in Figure 6 most

evident in southern Germany, are due to data gaps in the ASTER GED data and consistent across all variables. LST amplitude values range between 3 and 25 K (Figure 6b), with the lowest values where snow or ice cover is present all year round. High amplitude values are found in regions with seasonal snow cover that also heat up during the summer (Figure 7b). The phase of the harmonic oscillation (Figure 7c), shows spatial variations in seasonal shifts, which we report as the day of the year with the highest (peak) temperature in the annual LST cycle. The phase values display an altitudinal gradient (Figure 6c) with a

slight aspect dependence (Figure 7c).



**Figure 6. Landsat land surface temperature (LST) time series derived (a) mean annual LST (MALST), (b) LST amplitude and (c) phase of the harmonic oscillation and (d) LST trend across Switzerland and adjacent areas.**




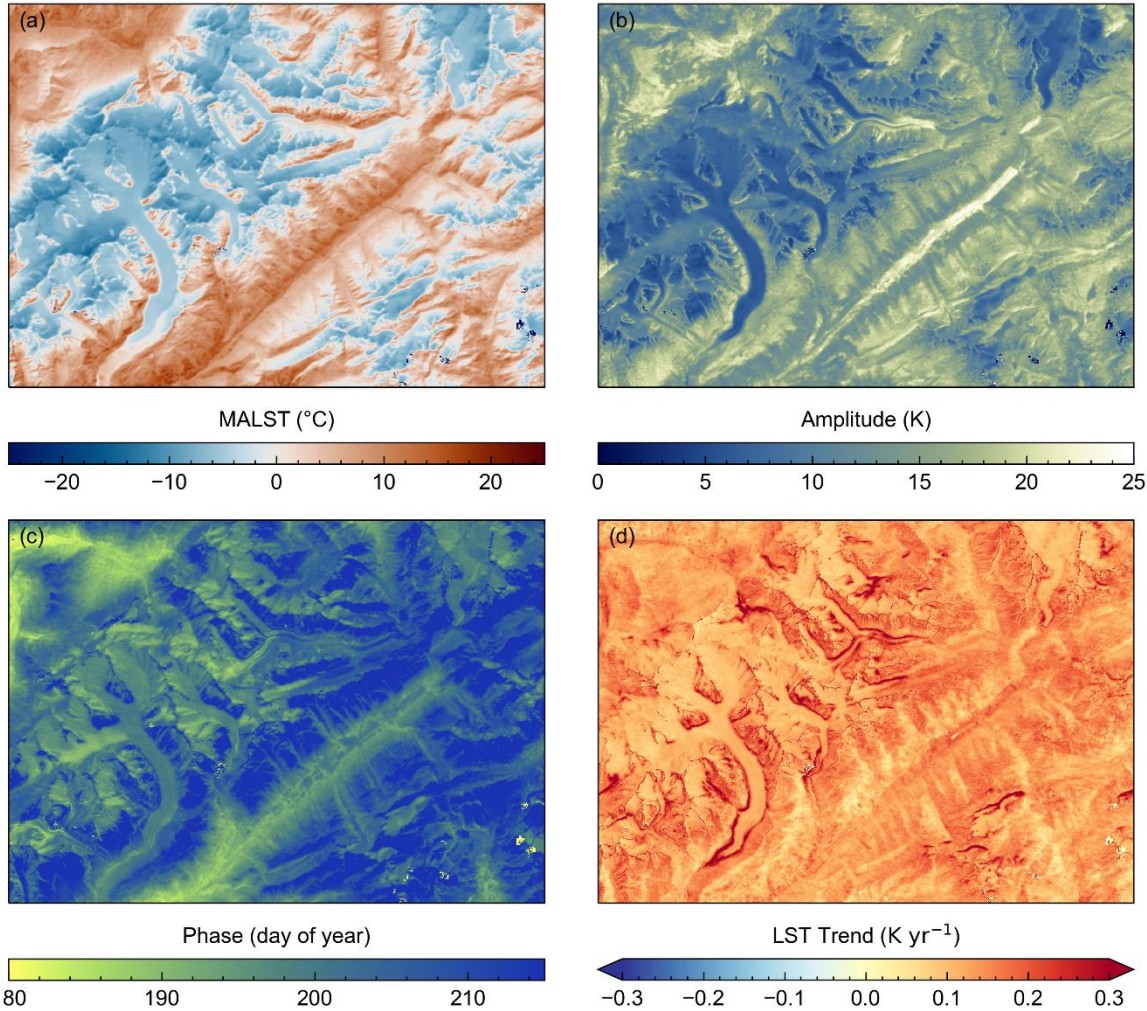

**Figure 7. Landsat land surface temperature (LST) time series derived (a) mean annual LST (MALST), (b) LST amplitude, (c) phase of the harmonic oscillation and (d) LST trend, across the upper Rhone Valley.**


Averaged across the entire study area, the mean LST trend is 0.14 K yr$^{-1}$ with the 5th and 95th percentile of 0.08 K yr$^{-1}$ and 0.21 K yr$^{-1}$, respectively. (Figure 6d). Areas with high population density often appear to exhibit trend values exceeding 0.2 K yr$^{-1}$. Notably, the highest trend values are observed in areas where retreating glaciers expose sediment or bedrock (Figure 7d). Compared to the MALST, LST amplitude and the phase of the harmonic oscillation, the LST trend values display more

artefacts. Subtle but systematic across-track jumps in LST trends are visible in the northeast of the map in Figure 6d. These artefacts align with the Landsat orbit and variations in the number of observations due to overlapping scenes from adjacent orbital tracks (Figure C1, see supplementary material). Similarly, the post-2003 Landsat LE07 scan line corrector failure induces across-track stripes in the number of LST observations that also appear in some parts of the LST trend maps (only



faintly visible on some glacier surfaces in Figure 7d). These patterns in LST trend values are consistent with the sensitivity to

record length we observed in our comparison of Landsat- and IMIS-derived LST trends (section 3.2). We further discuss this point in section 4.1. Finally, LST trends in the detailed map display an aspect dependency, with generally lower values at east-facing and higher values at west-facing slopes (Figure 7d). Regions with flat topography, as in the foreland, wide valleys, or lakes show more continuous trend values. We suspect that this effect is related to the shift towards later acquisition times and thus to variations in the solar zenith angle over the 38.5 years Landsat record. In the following section we examine this trend

bias in more detail using IMIS station data.

## 3.4 LST trend bias

To estimate the LST trend bias in flat to gently sloping terrain, we used LST data from the IMIS stations. The daily LST differences (ΔLST) at 9:29 h and 10:16 h UTC across all 119 IMIS stations, derived from linear interpolation of the 30-

minute interval raw data, show a bimodal distribution (Figure 8), which we separated using bimodal Gaussian regression. During melting periods, snow surfaces remain locked at the melting point and ΔLST values are essentially zero (blue curve). The remaining ΔLST values are normally distributed (red curve) with a mean ΔLST of 1.72 K and a standard deviation of 0.93 K. Over a 38.5-year period, this suggests an average LST trend bias of 0.045 K yr$^{-1}$ for flat to gently sloping terrain.

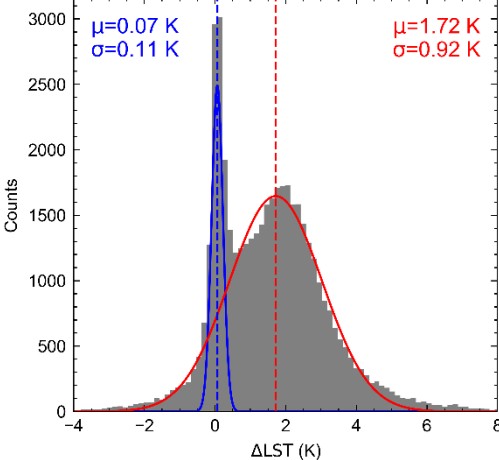

**Figure 8. Bimodal distribution of IMIS-derived land surface temperature differences (ΔLST) of daily LST interpolated at 9:29 h and 10:16 h. Mean (μ) and standard deviation (σ) were obtained from bimodal gaussian regression. Over the 38.5-year time period, a mean ΔLST of 1.72 K may thus explain 0.045 Kyr$^{-1}$ of the LST trend bias over flat and gentle sloping terrain where IMIS stations are typically located.**

The influence of topographic slope and aspect on the LST trends is shown by aggregated mean values for 2° slope

and 10° aspect bins in Figure 9c. For slope angles above ~10° LST trends are generally lower on east-facing slopes whereas they are higher on west-facing slopes. Mean LST trend values for slope angles above 75° are noisy due to very few samples (pixels) and have been excluded from analysis. We compared this pattern with modeled differences in incoming solar radiation

 

between 9:29 h and 10:16 h ($\Delta S_{in}$) for the 1st of all months of the year using terrain parameters from the DEM of our study area. In Figure 9d we show the pattern for July, which turned out to resemble the LST trend pattern the most, although

differences in $\Delta S_{in}$ patterns between May and September are generally small.

Overall, we find large similarities in the general pattern of how mean LST trends and $\Delta S_{in}$ vary with slope and aspect (Figure 9c, d; note that one colorbar is diverging while the other is continuous). Specifically, the cross sections shown for slope angles of 30° and 50° (Figure 9e, f), highlight the sinusoidal variation of LST trend and $\Delta S_{in}$ with aspect. We observe that the maximum and minimum values of LST trends for a given slope appear progressively translated to lower aspect values for

slopes >30°. This pattern is absent in the $\Delta S_{in}$ values. As expected, LST trend and $\Delta S_{in}$ variations with aspect are low for slope angles <10°. However, whereas the average $\Delta S_{in}$ value for any given slope and across all aspects is relatively similar, this is not the case for LST trends. There, we observe higher trend values for small slope angles when averaged across all aspects, compared to higher slope angles.

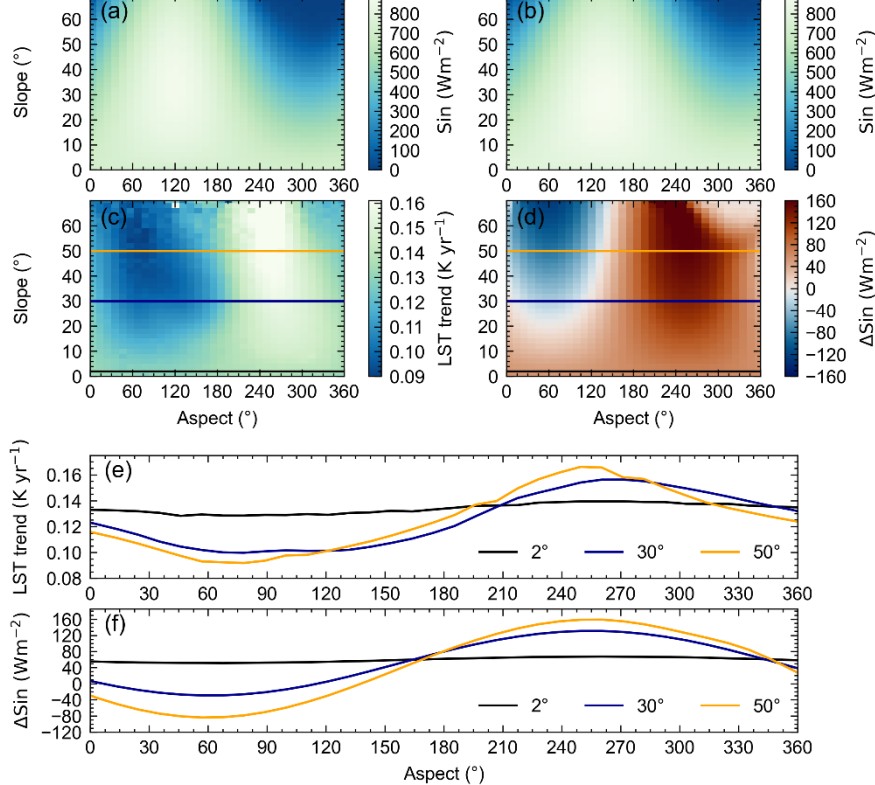

**Figure 9. Incoming shortwave radiation at 9:29 h (a) and 10:16 h (b), land surface temperature (LST) trend (c) and shortwave radiation difference between both times ($\Delta S_{in}$) (d) across Switzerland, excluding glaciers and all regions below 1700 m. Values are averaged for 2° slope and 15° aspect bins. Cross-sections of 30° and 50° slope angles show a similar sinusoidal pattern between mean LST trend (e) and mean $\Delta S_{in}$ (f), indicating LST trends biased by orbital drift.**



## 4 Discussion

### 4.1 Uncertainties related to LST and LST trends

Our comparison of Landsat-derived and in situ-measured IMIS LSTs has shown good agreement with a mean accuracy of 0.26 K for the combined Landsat sensors (Figure 4, Table 1). We observed no significant deviations in accuracy for the individual sensors, but the number of data points vary due to different temporal overlap of IMIS records and Landsat sensors. The slight positive bias of Landsat-derived LSTs greater than 0 °C, compared to those measured from the IMIS stations, is likely due to inaccurate IMIS LST data during snow-free conditions. The radiometric temperature measurements at the IMIS stations are based on a constant emissivity value of 0.98 for snow, resulting in biased temperatures for snow-free conditions. This explanation is consistent with greater accuracy at negative IMIS-derived LSTs, which often fall together with snow cover. The relatively large precision value of 4.69 K is likely in part due to the scatter around 0 °C, which is not necessarily a faulty or inaccurate measurement but rather caused by mixed-pixel effects due to the large resolution differences between IMIS and Landsat. During snowmelt periods, the IMIS sensor records ~0°C LST as long as snow persists under the sensor. Simultaneously, however, the larger footprint (60-100 m) of the Landsat measurement may record a mixed signal in the wider area around the IMIS station, potentially ranging from snow-free patches in sun-exposed areas to non-melting snow cover in shadows. By excluding data where IMIS LST is between -3.5 °C and +3.5 °C, the precision and uncertainty for L578 reduces to 4.37 and 4.38, respectively. Despite the relatively large uncertainty and a slight warm temperature bias, we find that the comparison of almost $4.5 \times 10^4$ LST measurements shows good agreement. We note, however, that the IMIS network's spatial distribution does not fully represent the topographic complexity encountered in high mountains, as the stations are mostly installed on flat to gentle sloping surfaces below 3000 m elevation.

The robustness of LST trends varies among Landsat sensors due to different temporal overlaps with the IMIS station data (Figure 2). Using LST data from all three sensors, the temporal overlap with IMIS LST data covers a record length of 22.6 years. Trends with such large temporal overlap are aligned well about the 1:1 line with a mean accuracy of -0.02 Kyr$^{-1}$, based on the residuals. However, this comparison is dominated by LE07, which has the longest overlap in the observation period (Figure 2). Although we are unable to evaluate LST trends from LT05 and LC08 based on long time series, our comparison together with the precious comparison of Landsat-derived and IMIS-derived LSTs for the different sensors provides confidence that LST trends derived from different Landsat sensors, spanning 38.5 years in total, are robust. Besides the record length, the total number of LST observations also plays an important role to derive robust LST trends. Although the Landsat archive covers four decades of LST observations, its temporal resolution of 16-day revisit interval is rather low. In addition, cloud cover renders many scenes unusable, highlighting the need for reliable cloud masking. This raises two problems, especially for mountainous terrain. First, frequent cloud cover leads to inevitable data gaps; and second, cloud detection algorithms are prone to failure over bright surfaces like snow and ice, which are common at high elevations. Our filter procedure, which is based on an initial LST model and thresholding the model-observation residuals in a second step, provides a way to detect unreasonably high or low LST values by taking the existing seasonal trend into account. We found



that this filter more often removes unreasonable cold LSTs, which are likely misclassified clouds, rather than warm LSTs. Yet,
it is also possible that the Landsat cloud flag might have classified bright surfaces as clouds, resulting in the possible removal

of valid LST observations. A robust and reliable cloud detection algorithm is currently the only practical way to minimize such
problems.

The number of observations in the LST time series vary not only due to clouds, but also due to other systematic
factors. Substantial spatial differences in LST counts arise from partial overlapping of adjacent Landsat paths (Figure C.1),
which tends to increase towards the poles. In our study area, these overlaps yield approximately twice as many observations

for a third of the area. Furthermore, the Landsat 7 scan line corrector failure further reduces data availability at smaller spatial
scales. MALST, amplitude and phase derived from LST time series seem to be generally unaffected by the variable number
of observations as no large-scale patterns following the mentioned limitations can be observed (Figure 6a, b, c). However, the
LST trend is more sensitive to the number of observations and subtle artefacts in some regions can be identified that align with
the flight path of the satellite (Figure 6d). In some regions faint stripes can be seen that correspond to the Landsat 7 scan line

failure and thus reduced data availability. We assessed the robustness of LST trend calculations with respect to the number of
observations through a systematic Monte Carlo simulation. By iteratively reducing the time series size (n=100) and performing
repeated trend analyses (1000 repetitions), we quantified the impact of data reduction on trend stability. Each value of the 1000
repetition was compared to the LST trend of the full time series (difference) and summarized as the mean and standard
deviation. We chose the Landsat LST time series at the IMIS location of OFE2, comprising 1009 observations with a LST

trend of 0.11 $Kyr^{-1}$, as an illustrative test site. The analysis revealed that although mean LST trend value remains stable across
sample sizes, the standard deviation, which represents the precision, varies more strongly. For common sample sizes of around
750 LST observations over the 38-year period, the 1-sigma value is 0.01.

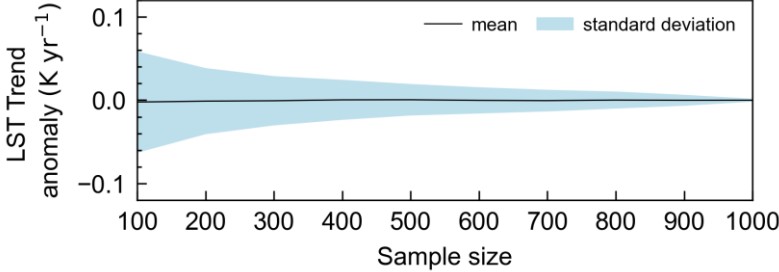

**Figure 10. Sensitivity Analysis of land surface temperature (LST) trend stability. LST trend anomaly shows the difference of LST**
**trend derived from full time series and repeated LST trend calculations (1000 repetitions) with iteratively reduced sample sizes**
**(n=100). Results are given as mean and standard deviation.**

**4.2 Clear-sky bias**

LST measurements based on thermal infrared remote sensing are biased towards clear-sky conditions (Ermida et al.,
2019). The effect of such a bias on LST trends has not yet received much attention. A recent study indicated no discernible
impact of clear-sky bias on LST trends (Good et al., 2022) by comparing satellite-derived LST with 2-meter air temperatures





under clear-sky and all-sky conditions. Further, Zhao et al. (2021) compared mean annual LST trends with trends in clear-sky day occurrence and did not identify a clear correlation for daytime LST but emphasized the challenges arising from changing surface conditions in the analysis. The Landsat data provides us with the timing of cloud cover and thus allows us to estimate the impact of cloud cover on LST trends at the IMIS locations. We compared IMIS LST trends derived during Landsat overpass

days at clear-sky days with IMIS LST trends derived during all Landsat overpass times, including clear sky and cloudy sky conditions. We found that on average LST trends during clear-sky conditions are 0.027 K yr$^{-1}$ warmer than during all-weather conditions (Figure 11). We note however that the spread in the data is relatively large and we are reluctant to generalize this finding. Nevertheless, this exercise suggests that for our study area an additional uncertainty of ~0.03 K yr$^{-1}$ is associated for comparison between clear-sky and all-weather conditions.

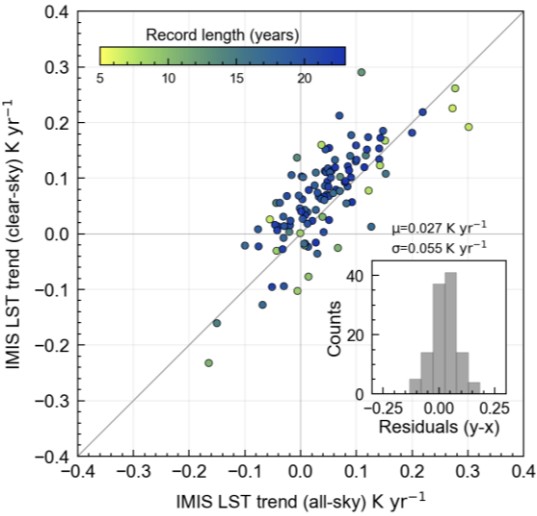


**Figure 11. Relationship between IMIS land surface temperature (LST) trends during clear-sky and during all-weather conditions. LST data were interpolated at Landsat overpass times.**

**4.3 LST trend bias due to changing acquisition times**

Our analysis of changes in IMIS LST during 9:29 h and 10:16 h UTC (Figure 8) and the spatial patterns of Landsat-

derived LST trends with slope and aspect (Figure 9) suggest the existence of an LST trend bias due to changing acquisition times. A linear fit of the acquisition times of all three sensors together does obviously not cover all the individual variations in orbit position. However, the close similarity of the slope and aspect dependency in LST trends and $\Delta S_{in}$ suggests that this approach appears to recover the first-order bias reasonably well. The dominant process that influences diurnal variations in LST during clear-sky conditions is the incoming solar radiation (Ghausi et al., 2023). Surfaces that are exposed to direct solar

radiation receive particularly high amounts of energy and are thus prone to heating up quickly during the morning hours, especially during the summer months. The additional radiation flux received during the 47-minute time window peaks for surfaces that are oriented orthogonal to the sun position, at an aspect value of approximately 130°, whereas the LST trend and $\Delta S_{in}$ peaks at approximately 75° and 255° respectively (Figure 9a, b). Instead, our results suggest that, rather than the total





amount of energy received, the spatial pattern in LST trend is more strongly controlled by the relative changes in direct solar

radiation ($\Delta S_{in}$) during the 47-minute time window, with positive and negative peaks at approximately westerly- and easterly-exposed surfaces, respectively. As a result, the greatest temperature changes occur where surfaces have an orientation that results in a switch between sun-exposure and shadow during the 47-minute time window. Observed differences in the slope-aspect dependence of $\Delta S_{in}$ and LST trends (Figure 9 a, b) are probably related to actual LST trends that are unrelated to slope and aspect.

Possibly the simplest way to deal with the LST trend bias due to changing acquisition times would be to choose an observation time period in which the orbital drift was minimal, such as 1998-2018, or by neglecting Landsat 5 data altogether and Landsat 7 after 2018 (Figure 2). We tested this shorter time period (Figure C2-4) and obtained LST trend values that were considerably noisier and more strongly affected by artefacts seemingly related to the number of observations (see section 0). We attribute this lower signal-to-noise ratio to the shorter observation time period, which also happened to be a limiting factor

in our comparison with IMIS-derived trend values (Figure 5). Previous studies concerned with the removal of the influence of orbital satellite drift on LST data – mostly for NOAA-AVHRR – employed different techniques (e.g., Julien and Sobrino, 2012) that are, however, difficult to implement for Landsat, due to substantially fewer observations and more heterogeneous terrain. In addition, correcting each observation to a consistent time before fitting Eq. 2 is prone to unquantified errors and spurious trends (Julien and Sobrino, 2012), and difficult to implement in GEE. We thus tested another possible approach,

which is to estimate the LST trend bias after the fitting, based on the strong observed terrain influence (Figure 9). This approach is probably less accurate as it neglects potential influences of different ground surface materials, but it is easier to implement. To do so, we first smoothed the map of mean $\Delta S_{in}$ for slope and aspect using local linear regression and normalized the values by the standard score. We then scaled the normalized model to approximate the observed LST trend pattern as a function of slope and aspect by least squares regression. Finally, we used the mean amount of surface warming (0.045 Kyr$^{-1}$) within the

47-minutes time window for flat and gentle sloping terrain from the IMIS stations (Figure 8) to align the model data for slope angles less than 10° (Figure 12).

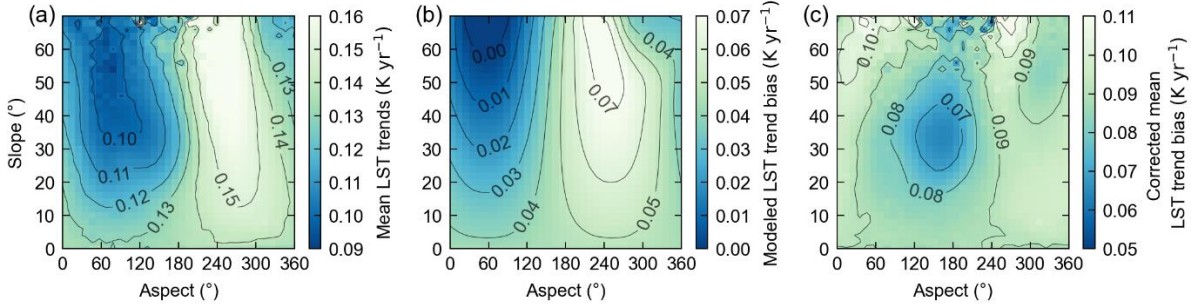

**Figure 12. Mean LST trends (a), modeled LST trend bias (b) and corrected mean LST trends (c) for 2° slope and 10° aspect angles.**

The modeled LST trend bias ranges between approximately 0 and 0.07 Kyr$^{-1}$, depending on slope and aspect. After

removing the estimated bias, the remaining LST trends (Figure 12c) still show some residual pattern that follows the





topography, with about 0.02 Kyr$^{-1}$ lower trend values centered on ~160° aspect and ~35° slope. The slope-aspect position of this residual LST trend feature is similar to the position of the highest $S_{in}$ values in Figure 9a & b. If there would be an additional influence of the additional $S_{in}$, received during the 47-minutes time period, we would expect LST trend values to be higher on surfaces approximately orthogonal to the sun vector, not lower, as suggested by the observations. Therefore, it presently

remains unclear, whether the residual LST trend feature is due to the LST trend bias and an inadequate correction, or possibly related to other processes. Applying the LST trend bias correction to the LST trends (Figure 13 derived from GEE results in overall lower trend values and less spatial differences in LST trends with respect to aspect. Further spatial variations that are still present after the bias correction appear to be related to differences as well as changes in land cover types and warrant further detailed inspection, which is beyond the scope of this study.


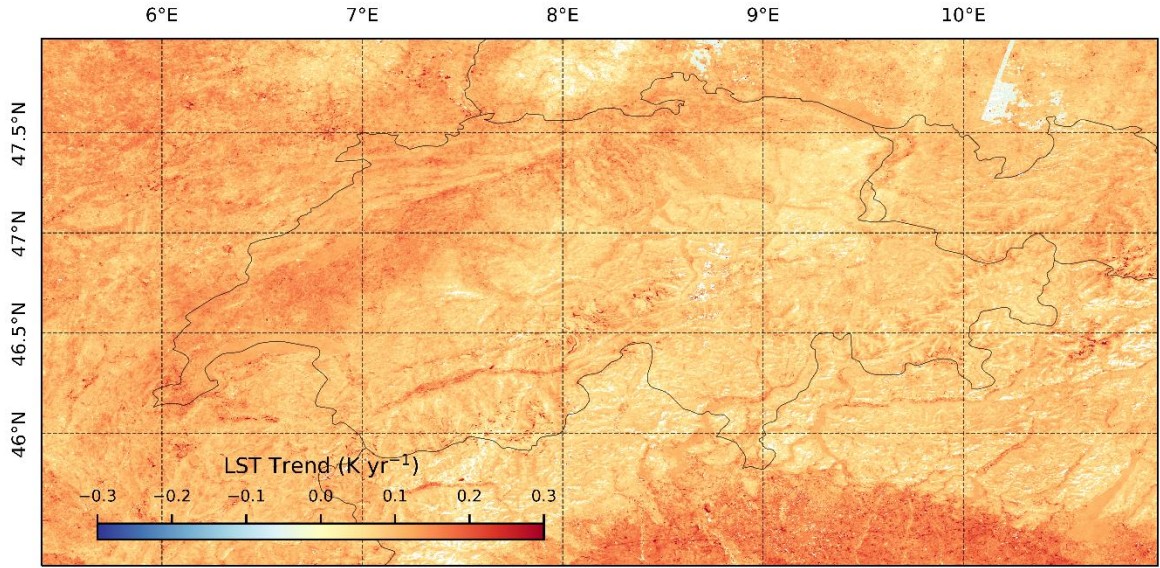

**Figure 13. Corrected land surface temperature (LST) trends of the Swiss Alps. Significance was estimated using a *t*-test and only significant (p < 0.05) LST trends are shown in the map.**

**4.4 Prospects for studying changes of the cryosphere**

Based on the corrected LST trend map, the spatially-averaged (±1σ) Landsat-derived clear-sky LST trend for all of Switzerland and for the time period 1984-2022, is 0.1 ± 0.05 K yr$^{-1}$. Insignificant (p>0.05) LST trends, determined by a t-test, were masked out and not considered. Most LST trend values range from 0.07 to 0.09 K yr$^{-1}$, with higher trends in populated valley bottoms like the Rhone Valley and lower trends over vegetated hillslopes at higher elevations (Figure 7). A detailed

analysis of LST trend variations with respect to different land cover types and properties as well as their change is beyond the scope of this study. However, we here briefly present examples of how changes in the mountain cryosphere map into spatial





patterns of LST trends at high spatial resolution. For instance, the rapid changes of mountain glaciers correlate well with patterns observed in the LST trends. Figure 14 shows as an example the Unteraar Glacier, where by far the highest LST trends occur along the glacier margin due to ice retreat and exposure of bedrock. Additionally, high LST trends are associated with

the expansion of supraglacial debris, which is well shown on the southern branch of the Unteraar Glacier, and the disappearance of clean ice in the lower few kilometers of the glacier. In contrast, LST trends are lower in magnitude and spatially more homogenous in the accumulation zone, which experiences minimal changes in surface type.

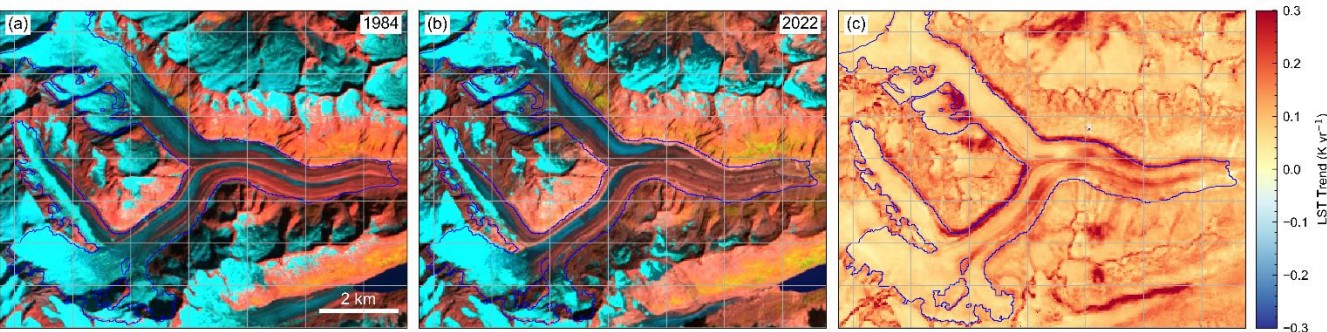

**Figure 14. Changes of the Unteraar Glacier, Switzerland, evidenced by late summer Landsat scenes from (a) 1984 and (b) 2022, and**
**by (c) land surface temperature (LST) trends. The satellite images show false color composites using the shortwave infrared 1, near infrared and red bands as red, green and blue channels. The blue line in all panels indicates the outline of the Unteraar Glacier based on the Randolph Glacier Inventory (RGI Consortium, 2017).**

How changes in snow cover influence LST trends would require a detailed analysis with respect to snow extent,
duration, depth and seasonality, which is beyond the scope of this study. However, in order to assess the first order sensitivity of LST trends to potential changes in snow cover, we spatially averaged LST trends for 100 m elevation bins and 1 °C MALST bins across the study area (Figure 1), excluding glaciers and glacier retreat zones (see section 2.4). Based on a previous global scale study of air temperatures we expect the highest positive temperature trends at altitudes where the MALST is between -10 and +5 °C, due to reduced snow cover and increased absorption of solar radiation (Pepin & Lundquist, 2008). Observed
mean LST trends at elevations where MALST is between -10 °C and 0 °C are among the highest trend values, consistent with an influence of snow cover on LST trends (Figure 15). In fact, LST trend magnitudes display a systematic pattern with MALST and elevation that merit more detailed examination. We note that MALST differences of up to ~20 K at similar elevation, are easily explained by different aspects, that is, exposure to the sun (see Figure 7a), which may coincide with different long-term trends in snow cover duration. Although dominantly negative mean annual snow depth trends, derived from the IMIS stations
by linear regression of annual mean snow depths further supports the effect of snow decline on LST trends, we did not find a clear correlation between LST trends and mean annual snow depth trends (Figure 14b). In addition, we do observe mostly positive trends in the number of snow-free days per year (Figure 15c), and these trends appear to increase in elevation. It is reasonable to assume that LST trends are higher where changes in snow cover are associated with more snow-free days, and that LST trends are likely smaller where snow depth declines but the surface remains nevertheless mostly snow covered,




similar as in glacier accumulation zones. However, a clear correlation between trends in the number of snow free days and
LST are not obvious, which could be related to the rather short record length of the IMIS stations and significant year-to-year
variability in snow depth and cover.

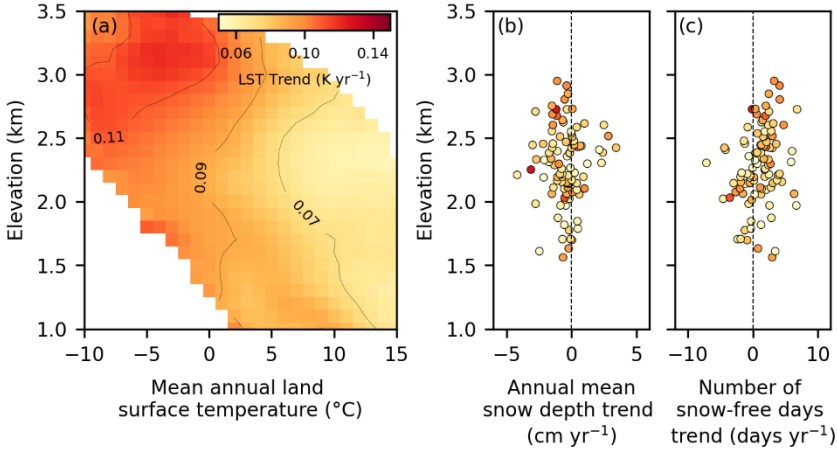

**Figure 15. Relationship between (a) mean land surface temperature (LST) trends for 100 m elevation bins and 1 °C mean annual**
**land surface temperature (MALST) bins, (b) annual mean snow depth trends and (c) trend in number of annual snow-free days at**
**the IMIS stations with more than 10 years record length.**

## 5 Conclusions

Our study has shown that Landsat-derived Land Surface Temperature (LST) since 1984 offer opportunities to study
the spatial variability of LST in complex topography at high spatial resolution. Our comparison with ground observations from
the IMIS network provides confidence in the remote sensing derived LST data and LST trends, despite challenges due to
differences in spatial resolution. The analysis of Landsat C2 LST time series, using harmonic regression including a linear
component, exploits the periodic nature of the intra-annual LST variation and yields maps of the mean annual LST (MALST),
the annual amplitude, the timing of the harmonic oscillation (phase), and the long-term LST trend. We observe reasonable
spatial patterns with elevation, slope and aspect that allow identifying the influence of surface orientation or type (e.g., glacier
surfaces) on annual LST variations. However, all LST time series components (i.e., MALST, amplitude, phase, trend)
presented in this study are based on LST at around ~10 h UTC and thus must be interpreted accordingly. In principle, the
Landsat archive provides a sufficiently long time series to obtain LST trends, as shown from our comparison with IMIS LST
data. LST trend values obtained from Landsat and the IMIS network converge for record lengths >15 years, whereas shorter
records exhibit considerably more noise. However, our analysis of the slope-aspect dependence of LST trends strongly suggests
that trend values are biased due to the long-term orbit changes that cause spurious LST trends. As orbit variations are not
uniform with time and sensor, a temporal coherence correction is challenging. Assuming a long-term linear change in
acquisition time, we have shown that the change in incident solar radiation can explain, at least in large parts, the spatial slope-

aspect patterns of Landsat derived 'apparent' LST trends. By modeling and removing the LST trend bias due to changing acquisition time, we obtain a spatially-averaged ($\pm 1\sigma$) Landsat-derived clear-sky LST trend for the time period 1984-2022 of

$0.1 \pm 0.05$ K yr$^{-1}$. The corrected LST trends respond to changes in the mountain cryosphere such as glacier retreat and debris cover evolution, snow decline and can potentially contribute to an improved prediction of permafrost temperatures, as surface temperatures propagate into greater depth. Further analysis is needed to disentangle the effect of land cover and land cover changes on the observed LST trends.

## 6 Data availability

Geotiff files of mean annual land surface temperature, amplitude, land surface temperature trend, RMSE and phase of the harmonic oscillation, together with the Google Earth Engine Python code are available from in the repository at https://dataservices.gfz-potsdam.de/panmetaworks/review/44fdf5c281ce564fc2d12f9e735781c3d01916eae3a3165d7d21c478b1682137/

## 7 Author contribution

**Deniz Gök**: Conceptualization, Methodology, Writing- Original draft, Writing- Review & Editing. **Dirk Scherler**: Conceptualization, Methodology, Writing- Review & Editing, Supervision. **Hendrik Wulf**: Methodology, Writing- Review & Editing.

## 8 Competing interests

The contact author has declared that none of the authors has any competing interests.

## 9 Funding

This research received funding from the European Research Council under the European Union's Horizon 2020 research and innovation programme under grant agreement 759639.

## 10 Acknowledgements

We are grateful to Noel Gorelick for advice with Google Earth Engine. During the preparation of this work the authors checked spelling and grammar using DeepL and ChatGPT to improve readability. After using these tools, the authors reviewed and edited the content and take full responsibility for the content of the publication.



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
