# Peer review of "Land surface temperature trends derived from Landsat imagery in the Swiss Alps"

_EGUsphere, 2024_

## Author Comment (AC1)

**Responses to Reviewer 1**

**egusphere-2024-1228: "Land surface temperature trends derived from Landsat imagery in the Swiss Alps" by Gök, et al. 2024**

Thank you kindly for taking the time to read our manuscript and for the constructive comments which helped us to improve it. In the following, we will address all comments point by point and suggest respective corrections. Please find our answers in blue font.

Surface warming trend detection based on land surface temperature estimations is a hot topic in recent years. This study provides a good attempt with the use of Landsat LST products, associated with good discussions. In general, the manuscript is well structured and the analysis is plentiful. Some additional revisions should be added before final acceptance.

1. L104: Section 0?There are serveral places with this number. Please

    Thank you for spotting the incorrect cross-reference. L104 refers to section 2.3 and L116 to section 2.4. We will change that accordingly.

2. Figure 2: I think the acquisition time here should be local solar time but not UTC time. Please check the details. Meanwhile, the symbols in the figure are not consistent.

    To our knowledge the timestamp in Landsat scenes is given in GMT, which can be seen equivalent to UTC. We therefore rather prefer to use the more general unit UTC time, which also is mostly used in remote sensing studies.

    Thank you for pointing out the incomplete legend in figure 2. We will add the symbol of excluded LE07 data and change the y-axis label.

[Figure]

3. Some basic introduction about the equipment for surface temperature monitoring should be added in this section.

    We will add to section 2.2.:

    "We used a subset of 119 stations (Table A.1) that provide radiometric surface temperature records in 30 min intervals. The IMIS stations measure radiometric surface temperature with an infrared sensor measuring in a wavelength range of 7 to 20 μm (David Liechti, person. communication, 2023)."

4. L165: I think the threshold should be set for one direction about the extreme low values, which should be affected by cloud cover.

    Yes, we agree that for identifying clouds the positive threshold is not necessary. However, we not only exclude remaining clouds, but potential outliers in general to ensure a robust trend

calculation. The positive threshold allows us to mask out extremely high temperatures caused by events like wildfires. We will add a remark that the positive threshold allows masking of wildfires, for example.

5. L224: Some more explanations about the high uncertainties appear at around 0-degree region should be added. Meanwhile, there should be no so many snow cover at this temperature range. The authors should confirm the impact factor.

   We agree and we will move the interpretation of the LST spread around 0°C from the results to the discussion section. That allows us to discuss the spread in more detail, including the calculation of precision and uncertainty when excluding data in the -3 to 3°C range.

6. Table 1: The metrics can be shown in Figure 4 and the table can be deleted.

   Although that is a potential option, we prefer to retain the tables in the manuscript to avoid overloading Figure 4 with additional numbers. The different types of metrics and thus the amount of numbers we wish to report is based on the recommendation in the "Land Surface Temperature Product Validation Best Practice Protocol" by Guillevic et al. (2017), where it is stated to include both mean and median-based error metrics to accurately report the accuracy, precision, and uncertainty of remote sensing LST products.

7. L246: From the simple comparison, it is hard to fully present the advantage of the LST data from LE07 because of the spatial discrepancy between satellite observation and field measurement. Meanwhile, there are gaps in LE07 data which may worsen the reliability of the trend assessment. I think it will be better to select the stations with good spatial representations to demonstrate the reliability of different products. Meanwhile, I think the trend analysis can be derived from all available LST observations but not from single one.

   We hope we correctly understood the concern. Indeed, the reliability and robustness of Landsat LST trends depend on both the total record length and the number of observations. The impact of record length on LST trends is addressed in Figure 5 a-c, where we compare LST trends derived from individual Landsat sensors with IMIS LST trends. We believe this comparison offers valuable insights for interpreting the combined record of LT05, LE07, and LC08 (Fig. 5 d). The similarity in the accuracy of LST trends derived from LE07 and the combined L578 suggests that LE07 LST data predominantly influences the derived trends in our comparison, as it overlaps most closely with the available IMIS data. We acknowledge that having LST station data covering the entire record length of LT05 and LC08 would allow us to further evaluate the impact of the different sensors on LST trends, irrespective of record length. Having said that, the spatial differences in the satellite-derived estimate and the field measurement remain a concern for all sensors. We address this issue in the discussion. At this point, we believe no changes are necessary.

8. Table 2: Similar as table 1, the metrics in table 2 can be moved to table 1.

   See our response to point 6.

9. L303: how to get this bias value? Please explain it in the text.

   Thank you for highlighting that the previous explanations were insufficient for understanding the estimation of the LST trend bias value. We will adjust the text to:

   "We pointed out in section 2.1 that Landsat acquisition times have changed between 1984 and 2022. Approximating this change by a linear model for the acquisition time, yields a time difference of 47 minutes over a period of 38.5 years (from 9:29 in 1984 to 10:16 in 2022; Figure 2). To estimate how much LST difference we would expect to result purely from this 47-minute

delay in image acquisition, we exploit the high temporal resolution IMIS data, by calculating for every day and every IMIS station the LST difference between 10:16 and 9.29. The daily LST differences ($\Delta$LST) show a bimodal distribution (Figure 8), which we separated using bimodal Gaussian regression. During melting periods, snow surfaces remain locked at the melting point and $\Delta$LST values are essentially zero (blue curve). The remaining $\Delta$LST values are normally distributed (red curve) with a mean $\Delta$LST of 1.72 K and a standard deviation of 0.93 K. Over a 38.5-year period, this suggests an average LST trend bias of 0.045 K yr$^{-1}$. However, the IMIS stations are located on flat to gently sloping terrain and the LST trend bias varies with topography."

10. How to consider the topographic influence on LST trends? It seems that here should be the variations of LST trends at different topographic conditions.

There are two important aspects considering the topographic influence on LST trends. First is the spatial variation of the LST trend bias caused by changes in Landsat acquisition times between 1984 and 2022. This bias varies spatially with topography because LST primarily depends on the amount of incoming shortwave radiation, which can change even within the 47 minutes of acquisition time differences. We covered this topic in sections 2.4, 3.4 and 4.3.

The second aspect concerns a potential spatial variation of LST trends with topography. After correcting for the above-mentioned trend bias with the approach described in section 4.3, we are unable to fully explain the residual variation of LST trends with slope and aspect. These residuals may result from an insufficient correction of the bias or represent a real trend signal that depends on topography. As there is evidence that snow cover and permafrost in the Alps also vary with topographic slope and aspect (e.g Kenner and Magnusson, 2016), changes in snow and ice cover could translate into the spatial variation in LST trends which we addressed in section 4.4. However, studying these effects in more detail would be beyond the scope of this study. Nevertheless, we added a sentence to the end of section 4.3 that points at the discussion in section 4.4 to make sure our discussion there does not go missing.

11. About the impact from the trend detection based on clear-sky observations, recent study has revealed this issue based on the comparison between the annual mean temperature from clear observations and all-weather observations from reconstruction works. Please refer to: https://doi.org/10.1109/TGRS.2024.3377670

Thank you, we will include the reference.

12. For the maps of the LST trend and other components, I think it will be better to show the Swiss Alps only with other countries removed out.

This certainly would be an option of displaying the results. Although our focus is indeed on the Swiss Alps, we think it is useful to show the larger-scale spatial patterns of LST trends, such as the transition from the alpine foreland into the Swiss high alpine regions. However, to better emphasise the study area, we will make the country outlines thicker.

[Figure]

13. About the uncertainty of the detected LST trends, although there are intercomparison with in situ observations, some additional comparison can be conducted with the observations from MODIS products. It will be much more helpful to identify the consistency or discrepancy.

We agree that a comparison of LST trends from other satellite sensors, such as MODIS, would be very interesting. However, in steep mountains, the coarse spatial resolution of 1000 m would average LST across steep spatial gradients, making a direct comparison between Landsat and MODIS LST trends difficult in such terrain. Implementing an appropriate correction to address this issue, would go beyond the scope of this study as the focus here is on the LST trend bias.

14. For the LST changes detected in this study, the discussion of the driving factors is not yet provided in current version. However, some necessary discussions should be added for this issue. Please refer to following articles: https://doi.org/10.1016/j.xinn.2024.100588, https://doi.org/10.1038/s41467-023-35799-4, https://doi.org/10.1016/j.rse.2018.06.010

Thank you very much for the references, these are great resources. As LST trends respond to climate change and land cover changes, the driving factors of LST trends over such large areas are very diverse. We are currently investigating this aspect in a follow-up study but think that, for the journal's focus on the cryosphere, a discussion on the relationship between land cover and LST trends would go beyond the scope of this study. With respect to changes in the cryosphere, we added section 4.4 which covers changes in snow cover and its impact on LST trends.

---

## Author Comment (AC2)

**Responses to Reviewer 2**

**egusphere-2024-1228: "Land surface temperature trends derived from Landsat imagery in the Swiss Alps" by Gök, et al. 2024**

Thank you kindly for taking the time to carefully read our manuscript and for the constructive comments which helped us to improve it. In the following, we will address all comments point by point and suggest respective corrections.

This study used Landsat-derived LST to analyze long-term LST trends in the Swiss Alps, referenced against ground observations from the IMIS network. Overall, this work is nicely done and offers valuable insights into identifying the origin of potential biases in Landsat-derived LST trends in mountainous terrain. I also find this paper is generally well written and structured. Below are some comments, primarily regarding the clarification of methodologies, which I hope the authors can address:

L55-L60: Please describe the spatial resolution of the thermal bands of AVHRR and Landsat.

We will add information on the spatial resolution and change the text to:

"As the robustness of trends increases with longer time series, LST records from the Advanced Very-High-Resolution Radiometer (AVHRR) **with 1000 m spatial resolution** and the Landsat Program (**60 – 120 m spatial resolution**), are particularly useful for this purpose (Prata, 1994; Gutman and Masek, 2012)."

L103: Change "artefacts" to "artifacts".

Thanks, done.

L104: Clarify what "section 0" refers to, similarly for L116.

Thanks. The cross-reference was incorrect. L104 refers to section 2.3 and L116 to section 2.4. We will change that accordingly.

L134: Ensure temperature units are consistent throughout the text.

Throughout the manuscript, we use degrees Celsius (°C) for absolute temperature values and Kelvin (K) for relative temperatures such as differences and rates. Figure 3 provides a good example for why we think it is useful to do so. Panel (a) shows LST in °C, which most people are used. Panel (b) shows residuals between modeled and observed LST in units K. Here, the unit of °C would erroneously raise the impression of reference to the absolute Celsius temperature scale.

To be clear about our usage, we will add to section 2.1:

"Throughout the study, we use unit degrees Celsius (°C) for absolute temperatures and Kelvin (K) for temperature differences and rates."

L156: Please briefly explain the filters used for masking clouds and duplicates.

Thanks for pointing out that more explanations are needed. As this point and the next are related to each other, we answer both comments here and suggest respective changes for a revised version of the manuscript.

We used two filters to prepare the LST time series for robust trend calculation (1) the removal of clouds and subsequently outliers from LST time series and (2) removal of multiple observations from the same day.

(1) The first order cloud removal is based on the Landsat Pixel QA band. We masked pixels that are cloud-flagged with at least a medium confidence tag in the QA band. Undetected residual cloudy pixels were further removed by applying a threshold on the residuals between modelled and observed LST. We chose a large threshold of +/- 30 K to not interfere with the high natural variability of LST and refer to previous work of Fu et al. (2014). Although most outliers in the LST time series can be attributed to undetected clouds (very low temperatures), we can't exclude very high-temperature outliers, by for example wildfires, that cause anomalous LST observations. Therefore, we applied the threshold for both positive and negative deviations. We will add a remark that the positive threshold allows masking of wildfires, for example.

(2) Landsat scenes are overlapping along- and across orbit track with the degree of overlap depending on the latitude. The across track overlap increases the number of observations from different days and is thus beneficial for LST trend calculation (see supplement figure C1). The along track overlap causes multiple LST observations of the same day which potentially can introduce a bias in the LST trends. To find along track overlap regions we implemented a filter for images in the same path, differing by less than 100 seconds. This approach allows to find each image's next temporal neighbor. From each select image pair we masked the overlap region in the neighbor the image.

We will change the respective section in the text to:

"Prior to fitting Eq. 2 to the Landsat LST data, we implemented filters to mask (1) duplicate LST observations with the same date that result from along-track overlapping Landsat scenes, and (2) cloud-contaminated pixels. **The along-track duplicates were removed by creating image pairs of each Landsat scene and its temporal neighbour in the same path and masking the overlapping region of the adjacent scene. The pairs of subsequent Landsat scenes were identified by a difference in acquisition time of less than 100 seconds which is small enough to only select the directly following scene.** Cloud masking was done using the Landsat C2 Pixel Quality Assessment Band (QA) cloud flag **with at least medium confidence** (Dwyer et al., 2018; Zhu and Woodcock, 2012)."

L165: Explain how the specific threshold is determined. It is unclear if applying an upper threshold of +30 K makes sense when trying to find cold extremes caused by undetected clouds.

See the previous answer.

L228-L229: The values of metrics do not match those shown in Figure 4 and Table 1. Please verify and correct them.

Thanks for pointing that out. We corrected the numbers in the text and checked the entire text to make sure no other issues exist.

Figure 5: You mention a total of 119 stations providing surface temperature observations, but only 115 are included in Figure 5d. Does this mean the remaining four stations had short time series and were excluded from the trend analysis? However, Figure 1 suggests all stations used should have consistent records for at least five years, please clarify. Additionally, the overall trends across stations derived from Landsat and IMIS LST should be given and compared.

Indeed, for the LST trend comparison we used LST data from only 115 IMIS stations that have sufficient long-time series (Figure 5). We added this to the caption of the figure. However, for the direct

comparison of IMIS and Landsat LST, the record length is not important, and we can make use of all available LST data. Therefore Figure 4 contains LST data from all 119 stations. We note that we do provide all LST trends from IMIS stations and the LST trends from the corresponding Landsat observations in supplement file A. A direct comparison between the trend values is possible, but complicated from the fact that they cover different and only partly overlapping time periods, the Landsat trends are affected by the bias we report, and the Landsat data is only from clear-sky days. Disentangling these different factors is what we attempt throughout our study.

L244-L245: While suggesting that LE07 is the most robust, it would be useful to see its distribution when restricting the record length to be comparable to LT05/LC08. This will help understand the impact of temporal overlap on the residuals.

We agree that further analysis on the residuals could reveal more insights on the impact of each sensor on the LST and LST trend measurements. However, here we focus on the impact of record length on LST trends. Differences in accuracy and precision for the individual sensors are given by the LST comparison in Figure 4.

L298-303: Does the ΔLST here represent the trend fitted by IMIS LST at 9:29 minus the trend at 10:16? If so, I assume you are explaining the LST trend bias caused by different acquisition times. Please ensure clarity. Also, explain how the average trend bias is calculated from the ΔLST.

Thank you for highlighting that the previous explanations were insufficient for understanding the estimation of the LST trend bias value. We will adjust the text to:

"We pointed out in section 2.1 that Landsat acquisition times have changed between 1984 and 2022. Approximating this change by a linear model for the acquisition time, yields a time difference of 47 minutes over a period of 38.5 years (from 9:29 in 1984 to 10:16 in 2022; Figure 2). To estimate how much LST difference we would expect to result purely from this 47-minute delay in image acquisition, we exploit the high temporal resolution IMIS data, by calculating for every day and every IMIS station the LST difference between 10:16 and 9.29. The daily LST differences (ΔLST) show a bimodal distribution (Figure 8), which we separated using bimodal Gaussian regression. During melting periods, snow surfaces remain locked at the melting point and ΔLST values are essentially zero (blue curve). The remaining ΔLST values are normally distributed (red curve) with a mean ΔLST of 1.72 K and a standard deviation of 0.93 K. Over a 38.5-year period, this suggests an average LST trend bias of 0.045 K yr$^{-1}$. However, the IMIS stations are located on flat to gently sloping terrain and the LST trend bias varies with topography"

L318-320: This statement is unclear; please elaborate on its significance.

Thank you for pointing out the unclarity. We reconsidered the statement and decided to remove the sentence in the revised version.

L349-355: Although mentioned in the conclusion, it is helpful to emphasize that the Landsat trends can be less reliable when the data period being examined is relatively short (e.g., less than 10-15 years).

Yes, good point, we will emphasize the effect of a shorter time series on LST trend variability and change the text accordingly:

"Trends with such large temporal overlap are aligned well about the 1:1 line with a mean accuracy of -0.02 Kyr$^{-1}$, while record lengths < 15 years show significantly more variability. However, long record comparison is dominated by LE07, which has the most overlap in the observation period (**Fehler! Verweisquelle konnte nicht gefunden werden.**)."

L413: It appears the LST trend peaks for an aspect of ~255° (Figure 9c?). Please verify it.

Yes, correct. We were referring to minimum and maximum values of LST trends and ΔSin found at ~75° and ~255°C. However, we think at this point it is sufficient to mention only the maximum value and will change the sentence to:

"The additional radiation flux received during the 47-minute time window peaks for surfaces that are oriented orthogonal to the sun position, at an aspect value of approximately 130°, whereas the LST trend and ΔSin peaks at approximately 255° (Figure 9)."

---

## Referee Report (RR1)

The authors have appropriately addressed my previous comments, which has improved the manuscript significantly with enhanced clarity. I recommend accepting the manuscript for publication after the following minor corrections:

L313, Should be "9:29".
L319, The mean $\Delta$LST should be calculated as the average of all values, including the melting ones (though close to zero). Thus, strictly speaking, the mean trend bias should be 1.72 K / 38.5 yr = 0.045 K yr$^{-1}$.

---

## Author Response (AR2)

**Responses to Reviewer 1**

**egusphere-2024-1228: "Land surface temperature trends derived from Landsat imagery in the Swiss Alps" by Gök, et al. 2024**

Thank you kindly for taking the time to read our manuscript again and for the constructive comments which helped us to improve it.

1) L313, Should be "9:29".
   Done.

2) L319, The mean ΔLST should be calculated as the average of all values, including the melting ones (though close to zero). Thus, strictly speaking, the mean trend bias should be 1.72 K / 38.5 yr = 0.045 K yr$^{-1}$.

   We adjusted the ΔLST value, which is now 1.55 K. This new value leads to an LST trend bias of 0.04 Kyr$^{-1}$. We recalculated the spatial variability of the LST trend bias based on the new ΔLST value and updated figures and text accordingly.

   Thank you for pointing that out.